# Rehabilitation Nurse’s Perspective on Transitional Care: An Online Focus Group

**DOI:** 10.3390/jpm12040582

**Published:** 2022-04-05

**Authors:** Rita Pedrosa, Óscar Ferreira, Cristina Lavareda Baixinho

**Affiliations:** 1Nursing School of Lisbon, Nursing Research, Innovation and Development Centre of Lisbon (CIDNUR), 1600-090 Lisbon, Portugal; oferrera@esel.pt (Ó.F.); crbaixinho@esel.pt (C.L.B.); 2Center for Innovative Care and Health Technology (ciTechCare), Polytechnic of Leiria, 2410-541 Leiria, Portugal

**Keywords:** advanced nursing practice, rehabilitation nursing, transitional care, hospital discharge, continuity of care

## Abstract

The increasing incidence of chronic and dependence leads to the need for hospitalization and adaptation in the process of returning home, as well as transition between care levels to ensure continuity of care. The World Health Organization has been warning about this problem since 2016, and consider reorganizing the care model as one of the solutions. The present study aimed to analyse the nurses’ perspective on transitional care for dependent people with rehabilitation care needs after hospital discharge. Methods: A focus *group* was developed with the participation of Rehabilitation Nurses from the hospital and community context, and content analysis was defined a *posteriori*. Results: From the content analysis emerged four related categories: promotion of continuity of care, nurse of advanced practice as a care manager, capacitation of the person and caregiver, and promotion of the care coordination. Conclusions: The present study allowed the strategies identification that minimize fragmentation risk of care and promote the person participation in transitional care. Ensuring transitional care is imperative to increase the quality of care, the satisfaction of professionals, clients, and the development of a system of sustainable health.

## 1. Introduction

The population aging associated with the increased prevalence of chronic diseases is a current problem and to which health systems seek to adapt responses through the development of strategies that provide better health outcomes and reduce the costs associated with the sector.

According to the World Health Organization (WHO), health systems have the responsibility to improve the population health and protect them from the financial costs associated with the disease situation, as well as to treat them with dignity. This would be made possible through increasing the capacity and quality of care delivery, ensuring coverage for the whole population, and ensuring that these services remain accessible to all [1]. 

Continuity and coordination of care are broad, interconnected concepts that can even overlap, and that contribute significantly to the way people experience their experiences in the health sector [1,2,3]. According to the WHO, there are eight priority practices that should be implemented at different levels of the health system and care, which ensure their continuity: (1) continuity (in the relationship) with primary health care professionals; (2) shared care and decision-making; (3) case management for people with complex needs; (4) services defined or with a single access point; (5) transitional or intermediate care; (6) integral service along the entire process; (7) technology to support continuity and coordination; (8) building the working capacity [1].

Transitional care refers to the care provision during the transition from hospital to home, from home to hospital and from illness or injury to recovery and independence—they can, in fact, involve the person’s reintegration into their employment and broader social role, or support the transition to palliative and end-of-life care. This holistic and biopsychosocial approach to transitional care should be culturally sensitive and involve the family, caregivers, employer and local community [1].

Countries with a health system based on Primary Health Care have a population with better health outcomes, lower rates of potentially avoidable hospitalizations, have lower socio-economic inequalities in reported health status and unmet needs. These outcomes are particularly relevant when interpreted in the face of an ageing population, and the consequent profile of the elderly (mostly users of primary health care and with frequent transition between levels of care): the integration and continuity of care are central and should provide the person, depending on their individual situation, access to the type and intensity of care they actually need, in the proper time and place [4,5].

The articulation between care levels requires a multi and interdisciplinary team that ensures quality and safety, avoiding the decline of functionality in the post-discharge period and unnecessary rehospitalisations due to foreseeable risks and complications. Ensuring a safe transition from the hospital to the community is, for this reason, an appropriate strategy to be followed by the different health services, by the potential in promoting autonomy and independence for self-care [6,7,8]. Nevertheless, transition between care levels often cannot be planned, which leads to consequences in the preparation of patients and caretakers and contributes to hospital readmissions. The reasons for readmission vary, but one of the most important components are assessment of community resources and whether they can fit patients’ needs [7,8,9]. A French study estimated that, for patients 75 years old or older, hospital readmission rate within 30 days after discharge was 14%, and according to the researchers one quarter of these events was preventable [7].

There are few studies on transitional care to support care continuity in the rehabilitation context and they have not systematized professionals’ interventions, which impacts the development of health policies oriented toward transitional rehabilitation care [9].

The present study aimed to analyse the nurses’ perspective on transitional care for dependent people with rehabilitation care needs after hospital discharge.

## 2. Materials and Methods

### 2.1. Study Design

This was a qualitative, descriptive and exploratory study to answer the following research question: “How rehabilitation nurses see the needs and organization of transitional care?” The study protocol consisted of five phases: planning, preparation, moderation, data analysis, and dissemination of results [10,11].

The *focus group* (FG) is a research method that aims to collect qualitative data from a group of people (between 4 and 12 participants) through their interaction and discussion on a topic presented by the researcher. The implementation of this method of data collection allows a greater intervention of participants on topics defined in a script, which can be carried out at different moments of the research process [10,11].

### 2.2. Participants

In the present study, the participants were defined as the following inclusion criteria: rehabilitation nurses, with clinical experience in rehabilitation care of dependent older persons’ with need of rehabilitation nursing care and with experience in transitional care programmes.

Given the nature of transitional care that starts at the hospital and continues after returning home, the option was to have 50% of participants with clinical practice in hospital and 50% with clinical practice in home care, integrated into primary health care. In accordance with the literature review that assume that transitional care and the management of the transition from hospital to home can happen in three stages: before the person leaves the hospital, at hospital discharge, or within 48 h to 30 days after discharge [1,2,3,7,9].

The number of participants was established a priori according to the ideal number of participants defined by the same authors, for online FG [10].

### 2.3. Data Collection

The literature review was crucial for structuring the interview script, which was organized around the follow stimulus questions: What are the transitional care needs presented by dependent older persons and their caregivers? What difficulties do they experience in ensuring continuity of care between hospital and community? How do you think transitional care can be planned to ensure continuity of rehabilitation care?

In the e-mail sent to nurses and at the beginning of the focus group, participants were informed about the aim of the study, the estimated duration (90 min), avoiding early dropouts at the start of the group discussions [11], and also about the presence and identification of the moderator and co-moderator.

The role of the moderator was to support the group in exploring the topic, and introducing new insights that may arise [11]. The stimulus questions and the work of the moderator were previously reviewed by the team to ensure the necessary moderation skills, group dynamics, and control of possible critical elements to ensure a successful FG. The choice of a co-moderator was in line with the recommendations of the authors, and was designed to increase the rigor of the process. His primary goal was to help the moderator manage the recording equipment, be aware of the conditions and logistics of the physical setting, respond to unexpected interruptions, and take notes on the group discussion [11].

Data collection was carried out online in February 2021, through the Zoom^®^ platform, with an approximate duration of 90 min. It was moderated by two impartial and experienced researchers.

### 2.4. Data Analysis

The recording was watched twice before being transcribed by one of the researchers present in the FGs so they could “visualize” what had occurred in the group.

After transcription, content analysis was performed according to Bardin [12], supported by webQDA software^®^.

It was conducted through the construction of categories of analysis a posteriori valuing and interpreting the information shared by the participants. Coding was carried out by the researcher who transcribed the recordings of the FGs, and then validated by the research team. Representativeness, comprehensiveness, homogeneity, and relevance were ensured when the categories were defined. A code was assigned to each participant (P1, 2, 3 …).

The research phases were rigorously conducted and validated by the entire team so that the results accurately represented the participants’ experiences. After coding the findings, they were returned to the participants for their validation, ensuring the credibility of the study.

Confirmability was guaranteed by the communication that took place during the coding process, by the literature, among the team, and by an expert who evaluated the codes that emerged from the findings.

Transferability was demonstrated by the depth of the analysis, the methodological description, and presentation of the results, which increase the likelihood of the findings being significant in other similar contexts.

### 2.5. Ethical Considerations

Ethical approval was obtained from the Ethics Committee of the hospital Vila Franca de Xira—Portugal (Protocol number—HVFX2020).

Anonymity and confidentiality were ensured and the data were encoded, without identification of the source. During the development of the research work, there was no provision for damage to the study participants or costs arising from their participation.

## 3. Results

Through the defined criteria, the study was developed through the participation of six rehabilitation nurses, aged between 40 and 53 years (representing an average age of 46.5 years), being mostly female (83.3%). All participants had a professional experience of more than 10 years, three of the participants performed functions in the hospital unit, and three in Community Care units integrated in the geographical area covered by the hospital unit.

From the content analysis emerged four categories and sixteen subcategories (presented in Table 1), being: promotion of continuity of care (with 93 units of registration), nurse of advanced practice as care manager (with 41 units of registration), capacitation of the person and caregiver (with 34 units of registration), and promotion of the coordination of care (with 96 units) (Table 1).

### 3.1. Promoting Continuity of Care

The promotion of continuity of care should include actions aimed, overall, at improving the health care provided and satisfaction of the person/family, minimizing the fragmentation of care, avoiding readmissions and consequently contributing to a more sustainable health system. In this context, rehabilitation nurses feel that all people would benefit from rehabilitation care, including dependents with reduced recovery potential and who would benefit from passive mobilizations for their comfort and safety. The participants recognize that the rehabilitation nurse will permanently be an added value in the continuity of care process, namely in the continuity of information.

“*… even in a completely dependent person who has come to the Emergency by extreme weight loss, by poor general condition, by the appearance [of] pressure zones, he may even be very dependent on his Barthel scale, but maybe if it is in a situation like the one I am describing, it makes perfect sense to be a rehabilitation nurse to go to the house*”(*P5*);

“*A user with multiple goings to the Emergency Service, with an identified caregiver, is necessary a person with quality, sufficient qualification to intervene in the process of management of those care that is being implemented, because if the person is going several times to the Emergency Service, there is a plan that has to be stipulated and corrected what is being done wrong. In my opinion it is a Rehabilitation Nurse who has human, technical capacity, whatever it is to define and to see what is badly stratified there at that moment, in the patient itself*”(*P6*);

“*It would have to be the assessment that is made at the patient’s admission, the evaluation made at the time of discharge, the gains acquired during hospitalization, or not, and the plan, what was the plan that was instituted that patient, the identification, who was the caregiver identified and who was made the teachings, or not, could have been only the user*”(*P5*).

The intervention of the rehabilitation nurse, for the participants, is simultaneously relevant in facilitating the transition process of health disease and, simultaneously, in the transition to the role of caregiver. If, on the one hand, there is a need to readapt the person to the health event and his/her disability, on the other hand, the family often sees moments of tension and anxiety associated with the moment of transition:

“*… having to start a life over with the problem you have at that moment, don’t you? And with the difficulties they will encounter*”(*P3*);

“*What I believe is that, usually these are older people who have a partner, or a partner, also the same age and that instead of coming an old man or the old woman, comes a child or daughter-in-law, because only they can move, and it is about those who do the teachings, they are the ones who are really targeted in care, obviously a user when he returns to the home, who is there is the husband or wife who is the same age*”(*P4*).

Nevertheless, they reveal the existence of difficulty in accessing care (UR = 13) during the transition, associated with difficulty in articulation (UR = 31).

“*… we had no information about what had happened, whether or not they had been in the hospital, and what the diagnosis was, and how quickly we had to make a home visit*” (*P1*);

“*And you have no information, in hospital terms, that that user is already followed by us, that caregiver has been followed by us for a long time*.”(*P1*);

“*… he*
*comes to a place and we start from scratch, we start asking the same questions and teaching the same things, and in a way maybe different. It doesn’t make any sense, does it?*”(*P5*).

The participants recognize difficulty in access and articulation, arguing that the rehabilitation nurse should take the lead in the organizational process of continuity care even if it lacks homogeneity of procedures. Through the focus group, it was possible to interpret the understanding about the health–disease transition process, the need to adapt to the present; however, the participants did not contemplate the integration of their desires and objectives in the rehabilitation process.

### 3.2. Advanced Practice Nurse as Care Manager

The category of advanced practice nurse as care manager (UR = 41) emerged from the subcategories related to the gains associated with the implementation of structured projects. Advanced nursing, and specifically advanced practice nurses, have been increasingly explored topics in research, namely in the contribution of these professionals with a characteristic profile, specialized knowledge, complex decision-making skills, skills and competencies for practice, whose characteristics must adapt to the context in which it is inserted.

Rehabilitation Nurses recognize gains associated with the person under going for care:

“*From the safe transition, when we go, most of them already come in a much better process in terms of autonomy*.”(*P1*);

“*And this transition is seen, and is mirrored in gains in the health of users, of patients who go to the emergency department, particularly those chronic users in which they had a certain number of already high emergency episodes, and that the community is absorbing them after our referral and are doing an excellent job with them*.”(*P6*);

At the same time, gains associated with the family are identified by the participants:

“*It is an added value too, just to complement a little, families verbalize*”(*P2*);

“*… caregiver’s anxiety*”(*P1*);

“*… even for family members themselves, it holds family members accountable and you help us in a completely different way from what you have done until then*”(*P6*).

They also recognize gains associated with professionals:

“*… also when we enter their house… gives another chega, another lens, so to speak, because it remains a conductive wire*”(*P2*);

“*… success is of articulation between all and has resulted very well*”(*P1*).

From the content analysis of this category, it is possible to conclude that rehabilitation nurses share a vision about their advanced and specialized practice, namely their contribution to improving the quality and access to care, resulting in benefits for the person in care, the family, and the professionals involved.

As already mentioned above, despite the recognition of professionals regarding the integration and coordination of care, the bibliographic sample analyzed in the integrative review of the literature demonstrates that the experience of the person himself translates into a non-inclusive role both in decision-making and in hospital discharge planning, associating the experience of the family that mentions distancing from professionals during hospitalization with consequent lower confidence in the rehabilitation process.

### 3.3. Training of the Person and Caregiver

In a similar perspective, the category of training of the person and caregiver emerges (UR = 34). In this context, and according to the specific competencies of rehabilitation nurses, it is extremely important to have early intervention by the rehabilitation nurse in the preparation of discharge, with the training of the dependent person promoting their functional readaptation and also intervening with their caregiver.

In the first instance, rehabilitation nurses approach the transmission of information to the person and caregiver, emphasizing the same importance of the training and learning process and its transversal language, both in the hospital context and in a home context:

“*Despite believing that all the teachings that are done inhospital are certainly very well accomplished, but our reality when we arrive on the first day to the family, they do not know anything and are completely lost*”(*P3*);

“*In fact when the colleagues in the community come and approach this person, they already know what we have done and can have a thread here to continue working on that person and have positive results*”(*P6*).

With regard to the intervention of the rehabilitation nurse in the person in the community after hospital discharge, this begins after referral still in the hospital context (via email or telephone) in order to continue the care plan already established, reinforcing some information that is considered fundamental.

“*Whenever a user comes … and that in the discharge letter or in the email is already referred to rehabilitation nurse, we try to always be [the Rehabilitation Nurse] to make the first evaluation*”(*P3*);

“*… dependence on self-care, dependence on mobilization, the risk of pressure ulcer, the risk of fall, the need for respiratory rehabilitation, are all criteria that we all have to speak the same language, we hospital for community and vice versa, right? And vice versa, if this happens, if it already comes with this identification and with this reference, the identification of these needs of the person*.”(*P5*).

With regard to the family, which assumes the informal caregiver role, it emerges as a partner in rehabilitation nursing care in the home context, also lacking interventions by the rehabilitation nurse.

“*… it is up to the Rehabilitation Nurse to evaluate the situation and then, from there, to be able to establish a plan together with the caregiver in a way that brings benefits to this patient*”(*P6*);

“*After this identification of the caregiver, it will have to be taught, instructed and trained in the needs that have been identified to optimize the potential of that person, both the person and the caregiver*”(*P5*);

It is also stressed by one of the participants the possible need for intervention in the housing context so that transitional care can be guaranteed, namely:

“*The evaluation of architectural barriers, here already by the colleague who is in the community, can also be something that is important for this continuity for the community*”(*P5*).

The susceptibility and vulnerabilities of the person and family in a transitional situation are a life-cycle phenomena influenced by several factors and variables, and that can ensure adherence to the rehabilitation program. The rehabilitation nurse assumes a fundamental role in the training of the person and caregiver, in an individualized and objective intervention, mobilizing instruments that allow identifying results and translating health gains.

### 3.4. Promoting Care Coordination

The category that presents the greatest expressiveness in content analysis is the Promotion of care coordination, with 96 registration units.

Care coordination should involve integrating interventions between care levels, in line with tools aimed at optimising care planning, including the transmission of information and monitoring of information and the current care plan.

Through content analysis, it is possible to observe that rehabilitation nurses value the optimization of care delivery in order to maximize the quality of care provided, believing that the moments of recourse to health services can still be experienced in a more pleasant way, reducing the rehospitalizations and minimizing the tension currently experienced.

The gains from transitional care are therefore clear for rehabilitation nurses:

“*… by creating the basic conditions, fighting hard for this, we would be able to ensure that the safe transition had a key role in patient care. To any of them*.”(*P6*);

“*If this were the case, we would certainly avoid many readmissions*”(*P4*).

Nevertheless, the participants express their intention to intervene with the teams to ensure the coordination of care and knowledge of the reality of the care of each context, thus assuming a leadership role.

“*… we should all get a view of what this is about, and you what’s [d]here, it’s more of an approximation. We had already talked here a few years ago about regular meetings, formal meetings*.”(*P4*);

On the other hand, rehabilitation nurses identify difficulties and needs in the implementation of their rehabilitation programs. It is reported by several participants that there is an increasing identification of people with oncological, palliative, or mental pathology who need the intervention of the rehabilitation nurse, and that it is difficult to respond to these requests concomitantly by the asymmetry of available resources.

“*In n functions, ready… Wherever it is, it has n functions, and it goes there a arrive, here and there, to try to safeguard everything, but it is not enough*.”(*P4*);

“*And we have no possibility of allocating resources, we have no means to deal with solicitudes*”(*P2*);

“*Nothing, nothing, on the contrary, we have to adjust when they should be themselves [computer applications] responding to our needs*.”(*P4*).

They also state that it is not possible for them to provide exclusively specialized care in rehabilitation, accumulating interventions that could be developed by general care nurses, but there is, on the part of the context, the need to monetize home visits.

“*In the ideal world there were enough nurses to be able to mobilize. So, they would have to have very defined functions within multidisciplinary teams and work with the social worker, with nutritionist, with the psychologist, with the … I don’t know, with the physiatra, to make a reference to the Network [National Care Network]*”(*P5*).

Although the benefits of transitional care and the evidence of the role of leader of these professionals in the projects are clear to the participants, they are referred to as an opportunity to improve the development of guidelines for referencing the person to the community, communication between professionals (method and language) and peer training, ensuring, for example, the identification of the need for visit by the rehabilitation nurse, or even in the continuity of teaching and training programs with the person and family.

Compared to the results obtained through the analysis of the bibliographic sample, health professionals, despite recognizing the success of transitional care dependent on the reintegration of care fragments, did not identify their opportunities for improvement explicitly, highlighting exclusively the barriers to integration and coordination of care (e.g., the lack of time available from the team for this investment).

## 4. Discussion

The results obtained allow us to affirm that, although there is a transition between levels of care, transitional rehabilitation care does not always exist. This results in an increased risk of fragmented care and the absence of continuity of care increases the risk of adverse events and complications in the period after discharge, contributing to hospital readmissions, loss of quality of life and increased co-morbidity. It is clear, for this reason, the need for transitional care that provides a positive impact in promoting independence for self-care and functionality to avoid complications after hospital discharge, which is corroborated by the results of the focus group [13].

Published studies state that, in the process of transition of care, the person identifies as fundamental the recovery of their autonomy, learning about self-care, the relationship with caregivers and professionals and the involvement in the transition of care (and their planning). From the experience of this transition emerge six indicative themes, namely: the need of the person himself to become independent, to learn about self-care, the relationship of support with caregivers, the relationship with professionals, the search for information, and the discussion and negotiation of the transitional care plan [14].

The rehabilitation nurses included in the focus group recognize that knowing the reality of the care of each context would be a facilitating aspect for the preparation of the return to home.

Despite being one of the priority practices defined by the WHO since 2018, studies on the effectiveness of transitional care programs report inconsistent results, partly as a result of differences in the service of each country and in the characteristics of the world population. Nevertheless, few studies have evaluated the factors that affect the success of the implementation of transitional care, and in existing studies may have inflated the careful selection of people who could benefit from this type of care and reduce their efficiency and effectiveness of care [1].

Continuity and coordination of care have an extraordinary impact when interventions are an integral part of a comprehensive care model, defining primary care as a focus. The evidence suggests that the effective management of the hospital–home transition, in addition to other benefits, speeds up the functional recovery of the person, estimated in functional gains greater than or equal to 35% [1,15,16].

The moment of hospital discharge and return home is also a challenge for the family, as it will be the moment when self-care management will be their responsibility, in need of numerous adaptations to what they were assisting in the hospital context. Hospital discharge planning is therefore one of the strategies for action to optimize this transition, maximize adaptation and reduce the risk of hospital readmissions.

The focus group participants recognize this need and identify gains for the family by participating with the interventions of the rehabilitation nurse in terms of transitional care, reinforcing the idea that, for the transition of care to be successful, it is necessary to plan, prepare, education for the health of the individual and his/her family from the moment of his/her hospitalization [17,18]. However, the changes are not always addressed by health professionals with due relevance, which thus provides a fragmentation of care after discharge. Often, the guidelines for discharge are carried out in an automated and hasty manner, only provided at the time of discharge, without considering the conditions and needs of each client and his family. Even when discharge is properly prepared and the user and family feel confident, they can return home and still experience difficulties and uncertainties regarding their treatment and recovery [17].

Another theme that finds convergent points in scientific production and the focus group is the role of health professionals, over whom the challenge of interaction, coordination, and integration between caregivers at different levels of care, ensuring the planning of discharge and a subsequent safe follow-up, with the involvement of the person and his caregiver, prevails, at all stages of transitional care. The coordination of care should involve the integration of interventions and levels of care (vertical and horizontal integration), using specific mechanisms and instruments for care planning, including the transmission of information, monitoring of needs and therapeutic plans, with the aim of optimising the provision of continuous and comprehensive care, in due place and time [19].

Through the data obtained through the realization of the focus group, rehabilitation nurses condensed their intentions in the development of guidelines for referencing the person to the community, in the standardization of communication between professionals (method and language) and peer formation. They also mention, as a proposal, to improve the standardization of it systems so that it is possible not only to share information, as well as to monitor developments in the care process according to the same health indicators.

Health care needs have evolved and, as such, specialized nursing care, and specifically advanced practice nurses have emerged as an attempt to adapt responses to these needs. A nurse with advanced training is a duly accredited nurse who has acquired specialized knowledge, has a high decision-making capacity and clinical skills for advanced practice [20], as well as for the evaluation and management of care for chronically ill patients, for example after hospital discharge [21].

The qualified performance of nurses is recognized as fundamental for the realization of safe transitions, as well as contributing to the visibility and valorisation of nursing intervention. However, the nurses that participated on the FG state that it is not possible for them to provide exclusively specialized care in rehabilitation, accumulating interventions that could be developed by general care nurses.

Professionals also identifies other barriers to the integration and coordination of care and family involvement, such as the lack of team time for this investment, the complexity of the person’s health and their post-discharge care. Some studies also highlight the difficulties of communication and articulation between levels of care as an impediment to an integrated response to the needs of the population with complex health–disease problems [7,9,13,22].

There are some tools and interventions for transitional care, which enhance the participation of the person in his/her discharge planning and rehabilitation, namely: family meetings; preparation of discharge planning; existence of checklist respecting the needs of care identified by the person and caregiver, available community resources, the need for support products, among others; definition of a health education program; and home visits that may include the decision-making process of the person, involving the family and caregivers, facilitating participation in their daily life [23].

Participants advocate the need of strategies and health policies to increase the coordination and vertical integration of community and hospital care. This continues to prove to be a persistent challenge with repercussions already demonstrated in the tendency to reduce hospitalization time, home follow-up and, consequently, in reducing the rate of occupancy of hospital beds. Ensuring this coordination between levels of care, considering the risks at different stages of the cycle, and ensuring transitional care is imperative for the development of a more sustainable health system [24,25].

In other countries, the scientific evidence is clear with regard to successful interventions and implementation strategies for the transition of care, namely the integration of people with change of functionality in policy processes with a view to improving the responsiveness, efficiency and effectiveness, and sustainability of programs, strengthening self-determination and user satisfaction; the collection of statistical data on the development of health information systems, with the aim of supporting a political impulse, decision-making in policy reformulation and equitable allocation of resources; cross-sectorial coordination in the provision of rehabilitation care; and the establishment of a rehabilitation program aligned with pre-existing health programs, supporting its sustainability. It is emphasized as an action strategy, specifically, the development of a community-focused care program (primary health care), promoting home visits after clinical discharge, and the designation of a nurse case manager as the process leader [26,27].

The management and leadership of transitional care programs should undoubtedly be attributed to nurses of advanced nursing practice. The advanced practice nurse is often seen as the clinical specialist, with functions that include understanding and influence on management issues, policy development and clinical leadership. Although the core of the practice is based on advanced technologies, education and knowledge, being flexible according to the reality of each country, the distinction between advanced and generalist nursing practice is clear. Still, the nature of advanced nursing practice concerns a designated function focused on care delivery in the field of prevention and cure, including rehabilitation care and chronic disease management [20].

The leadership of transitional care programmes includes the perception of the rehabilitation nurses who constituted the focus group, who recognise their contribution to improving the quality and access to care [28], and in the gains associated with the person, family and professionals involved in the projects already implemented, namely autonomy at the time of return home, the reduction of the use of the emergency service, the minimization of the caregiver’s anxiety, and the articulation between rehabilitation nurses. In short, it is important to emphasize that, although it is imperative to monitor health policies in relation to the care needs of the Portuguese population, there will be no exclusive or unique approach that solves the necessary reform of health systems. In this sense, it is suggested that the general recommendations of the world organizations can be followed, flexibly adopting the practices that can increase the effectiveness of health spending and the efficiency of health systems, adapting to the socio-geopolitical context in which we are integrated.

The limitations of the present study are mainly due to the method itself. For the focus group method, the limitations also stem from the selection of the method, highlighting a small representative sample and limited to a geographical area.

It should be noted that no Portuguese studies on the theme under study have been identified, leading us to believe that the contribution of this study may foster interest in the area of study and development of further studies.

## 5. Conclusions

The present study allowed the identification of strategies that minimize fragmentation risk of care and promote the participation of the person in transitional care. Ensuring transitional care is imperative to increase the quality of care, the satisfaction of professionals, clients, and the development of a system of sustainable health.

At a macro level, even though the health system has tried to keep up with the evolution of the needs of the Portuguese population, there are still inefficiencies and gaps in the transition of care, often associated with communication, articulation and access to care, harming the general objective of a health system, promoting health care of higher quality, with better health outcomes and associated with a lower cost.

Nevertheless, the lack of formalization of the care transition process is a reality, lacking explicit national policies. The coordination and integration of community care with hospital care is a persistent challenge, demonstrating a tendency to reduce hospitalization time, home follow-up and, consequently, the rate of occupancy of hospital beds. Training, monitoring and coordination between levels of care needs to be ensured.

It is concluded that ensuring transitional care is imperative for the development of a sustainable health system, increasing the quality of care and the satisfaction of professionals and clients.

## Figures and Tables

**Table 1 jpm-12-00582-t001:** Corpus of the content analysis from the focus group, Lisbon, 2021.

Category	Subcategory	Registration Units
Promoting continuity of care	Continuity of care	38
	Access difficulties	13
	Difficulties in articulation	31
	Health–disease transition	2
	Transition to the role of caregiver	9
	Subtotal:	93
Advanced practice nurse as care manager	Gains with structured project—Person	22
	Gains with structured design—Family	4
	Gains with structured design—Professionals	15
	Subtotal:	41
Training of the person and caregiver	Information	6
	Rehabilitation nurse intervention—Person	16
	Rehabilitation nurse intervention—Family	11
	Rehabilitation nurse intervention—Home	1
	Subtotal:	34
Promoting care coordination	Gains with transitional care	5
	Needs felt by professionals	59
	Difficulties experienced by professionals	30
	Rehabilitation nurse intervention—Team	2
	Subtotal:	96
	TOTAL:	264

## Data Availability

Data are available upon request to the authors.

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
