# Peer review of "Rehabilitation Nurse’s Perspective on Transitional Care: An Online Focus Group"

_jpm, 2022, doi:10.3390/jpm12040582_

Round 1

Reviewer 1 Report

Research related to continuity of care is of interest to advance the application of evidence-based practices in relation to improving the transition of care.
However, despite the research topic is important, this manuscript has important limitations such as the research question is not clear, the objectives are not well defined, the selection of these participants is not justified, only six Rehabilitation Nurses were included in a single discussion group as data collection technique:three of the participants perform functions in the hospital unit and three in Community Care units integrated in the geographical area covered by the hospital unit. Incomplete reliability and validity in this qualitative research.

Author Response

We thank you for your contribution to the revision of our work. We selected 6 participants in line with the recommendations of other authors who note that in online focus groups are generally more effective with fewer participants than might be effective during in-person groups.

References:

Kite J, Phongsavan P. Insights for conducting real-time focus groups online using a web conferencing service. F1000Research. 2017 Feb 9;6:122–122.

Burton LJ, Bruening JE. Technology and Method Intersect in the Online Focus Group. Quest. 2003 Nov 1;55(4):315–27.

Tuttas CA. Lessons learned using Web conference technology for online focus group interviews. Qual Health Res. 2015;25(1):122–33.

Stewart K, Williams M. Researching online populations: the use of online focus groups for social research. Qual Res. 2005 Nov 1;5(4):395–416.

As the focus is on transitional care, the criterion was to have nurses from both the hospital and the community who provide care directly to people who, after hospitalisation, with need of home care, in order to have a perspective of how the planning and organisation of this care can be articulated.

We have clarified the objective and methodological procedures to make the method section and the understanding of the results clearer.

Reviewer 2 Report

Please find the review report attached. 

Author Response

We thank you for your careful consideration and review of the article and the suggestions made, which will be taken into consideration in the new version of the article.

  • Reference 11 is now reference number 12 and is from a book on content analysis, which is not available in electronic format.
  • We changed some paragraphs of the introduction and mentioned why this study was necessary (page 2, line 73-76).

  • We have changed the method section in an attempt to make the methodological procedures clearer and incorporated your suggestions that are shading in yellow.

  • In the results we have removed some ‘verbatim’ (participants speech), keeping those that best illustrate the researchers’ interpretation of the results. We chose to keep the table because it translates the body of the content analysis, so that the reader understands that each category emerged from the grouping of the subcategories that compose it. In relation to the numbers that appear are the registration units, i.e., the text segment that is the target of selection for analysis - we followed the method proposed by Bardin.

  • In the discussion we removed text whose idea was already contained in the introduction and augmented the discussion with the results.

  • In the conclusion, we removed the excess information and focus on the results of the study.

Round 2

Reviewer 1 Report

We thank the authors for the changes in the manuscript, improving its presentation and understanding, especially in the methodology section. In this section, the term cross-sectional design should be removed, as it is a qualitative study.

Author Response

Dear reviewer:

We thank you for your contribution and have introduced the amendment – page 2., line 82.

Thank you